# Exploring barriers to the adoption and utilization of improved latrine facilities in rural Ethiopia: An Integrated Behavioral Model for Water, Sanitation and Hygiene (IBM-WASH) approach

**Aiggan Tamene** [1] *, **Abel Afework** [2]

1 Environmental Health Unit, School of Public Health, College of Medicine and Health Sciences, Wachemo University, Hosaena, Ethiopia, 2 Infection Prevention and Patient Safety Unit, Dilla University Hospital, College of Medicine and Health Sciences, Dilla University, Dilla, Ethiopia

* apublic22@gmail.com

## Abstract

### Background

Even though evidence shows that access to and use of improved latrines is related to healthful families and the public, obstacles to the adoption and use of improved latrine facilities remain. Globally, not many inquiries appear to have been carried out to satisfactorily inform us regarding the multi-level barriers influencing the adoption and utilization of improved latrines facilities. Related studies in Ethiopia are even fewer.

### Methods

Two qualitative data gathering methods, viz., key informant interviews and focus group discussions, were employed to collect data for this study. A total of fifteen focus group discussions were conducted with members of the community in the rural Wonago district of Ethiopia. Similarly, ten key informant interviews were conducted with water, sanitation, and hygiene officers, and health extension workers responsible for coordinating sanitation and hygiene activities. Open code software 4.03 was used for thematic analysis.

### Result

Barriers to adoption and use of improved latrine facilities were categorized into Contextual factors (e.g. Gender, educational status, personal preference for using the field, limited space, population density, the status of land ownership), Psychosocial factors (Culture, beliefs, attitudes, and perceptions of minimal health threat from children's feces), and Technological factors (inconveniences in acquiring materials and cost of constructing a latrine).

### Conclusion

There are a series of multi-leveled barriers to the sustained adoption and use of latrines. Providing funding opportunities for the underprivileged and offering training on the

**Data Availability Statement:** The qualitative data sets generated and/or analyzed during the current study are not publicly available due to the data

containing information that could compromise research participant privacy and consent but are available from the corresponding author on reasonable request, and subject to approval from the School of public health, ethics and research committee at the University of Wachemo. Data requests can thus be sent to the School of Public Health, data access committee (akliluha@wcu.edu.et).

**Funding:** The author(s) received no specific funding for this work.

**Competing interests:** The authors have declared that no competing interests exist.

**Abbreviations:** CLTSH, Community-Led Total Sanitation and Hygiene program; FGD, Focus group discussion; HEP, Health Extension Program; HEW, Health Extension Worker; IBM-WASH, Integrated Behavioral Model for Water, Sanitation, and Hygiene interventions; IRB, Institutional Review Board; KII, Key informant interview; SDGs, Sustainable Development Goals; WASH, Water, Sanitation and Hygiene.

engineering skills of latrine construction at the community level based on the contextual soil circumstances could expand the latrine coverage and use. Similarly, taking into account the variability in motivations for adopting and using latrines among our study in Ethiopia and other studies, we implore public health experts to recognize behaviors and norms in their target communities in advance of implementing sanitation interventions.

## Introduction

Improved latrine facilities are excreta disposal facilities that can guarantee the hygienic separation of human excreta from human and insect contact [1]. Improved latrine facilities include private improved pit latrines (PIPL), private traditional pit latrines (PTPL) with slab and superstructure, composting toilets, and flush or pour-flush toilets linked to sewage systems and septic tanks; Unsanitary toilets (USTs) such as a flush or pour-flush toilets, pit latrines without superstructure, open pit, bucket latrines, hanging toilets, and open defecation (OD) are considered as unimproved typologies of latrines [2]. Access to and use of improved latrine facilities has wider implications that extend beyond the prevention of excreta-related infections [3]. Studies have revealed that access to these services not only enhances psycho-social welfare but has economic benefits as well [4].

The Human Right to Sanitation declaration stipulates that every nation must ensure physical and economic access to latrine facilities along with services that are reasonably priced, safe, hygienic, secure, and socially and culturally acceptable while providing privacy and ensuring dignity to its citizens [5]. However, despite the presence of such well-meaning policy outlines, to this day, 892 million people practice open field defecation globally whereas, the number of people who utilize unimproved latrine facilities has surpassed 856 million [6]. In both cases, the majority of those affected are the impoverished rural populace found in the least developed countries [7].

In the developing world, among other things, poor sanitation has become the foremost challenge for both the public health and economic systems [7]. In 2012, poor sanitation together with scarce water and hygiene services was to blame for 824,000 diarrheal deaths in low and middle-income countries, accounting for 58% of the total diarrheal deaths. Of these deaths, nearly 361,000 (over 1000 deaths per day) occurred in children below the age of five [8]. In 2015, the pooled global economic loss attributed to sanitation-related premature deaths, health care expenditure for treating sanitation correlated diseases, output lost due to illness, and time lost to access sanitation facilities was estimated to be 222.9 Billion Dollars [9].

Ethiopia succeeded greatly in the reduction of open defecation practices from 80% in 2000 to 27% in 2015 [10]. The Government's implementation of the Community-led Total Sanitation and Hygiene Approach (CLTSH), a participatory behavioral change approach to rural sanitation that was formally adopted by the Federal Ministry of Health (FMoH) in 2011 and rolled out across the country through the Health Extension Program, a program launched 15 years ago focusing primarily on improving hygiene and sanitation at the household level, has largely allowed this significant achievement [11].

However, for latrine interventions to result in meaningful improvements to public health, behaviors and technologies ought to be adopted and sustained at scale over time, but evidence of sustained adoption of new practices in Ethiopia is mixed. A recent study has revealed that Ethiopia's trend of open defecation is growing [12]. Unimproved toilet facilities are used by 59% of rural households, according to the 2016 Ethiopian Demographic and Health Survey

report. One in every three households has no toilet facilities in the country and 60–80% of all communicable diseases, as well as 50% of the undernutrition befalling the country, are attributed to poor sanitation, hygiene, and lack of access to safe water sources [13].

In the Southern Nations, Nationalities, and Peoples' Region of Ethiopia, strong district-level inequalities exist in the coverage of improved latrine installations [14]. While some districts have reported substantial improvements in behavioral outcomes, others have stated an attenuation of initially improved practices [8]. Compared to the national figures, the Wonago district in the southern region has not kept pace with the increase in latrine adoption and utilization. 33.5% of the populace in Wonago district practices open defecation. The combined results of open defecation-free slippage and rapid population growth are the cause of such somber statistics [12].

Within 1 and 2 years of Open defecation free (ODF) certification, most of Ethiopia's ODF villages slip back [14]. This is partly due to the health system's relative weakness in monitoring progress and helping people move up the sanitation ladder, where they progress from open defecation to the use of simple latrines, to the use of more improved latrine options. Progress has also been affected by weak engagement from health extension staff and facilitators. Other factors leading to the slippage are the induction of CLTSH without sufficient planning and the lack of successful social mobilization methods [15].

In many of the prior studies conducted in Ethiopia and other African nations having a larger family, an advanced education, and being male were associated with the increased adoption and utilization of latrines [16–19]. Also, Qualitative studies on sanitation have identified the un-affordability of building materials [20] and the lack of awareness of the health risks associated with open defecation as a hindrance to the adoption of sanitation facilities [21]. Nevertheless, the majority of the studies focused on a single-level analysis [22–24].

Researchers, in the past, have affirmed that interventions that target multiple levels produce more sustainable transformations than individual-based interventions from a single-level analysis [4, 11]. Therefore, to find sustainable solutions and make headway in extenuating the problems regarding the adoption and utilization of improved sanitation facilities, it is important to fully understand the factors that are hindering the adoption and consistent use of such facilities.

To genuinely shed light on the multi-leveled nature of the present research, the Integrated Behavioral Model for Water, Sanitation, and Hygiene interventions (IBM-WASH) approach was found to be most effectual. The IBM-WASH, a theoretical model specially developed to understand water, sanitation, and hygiene interventions and behaviors, is widely accepted as providing an across-the-board methodology for analyzing and addressing multiple levels of influences. The model identifies three dimensions of water, sanitation, and hygiene-related behavior (contextual, psychosocial, technological) at five separate societal, community, household, individual, and habitual levels [25].

The contextual dimension encompasses the character of the individual, setting, and environment that are outside of the scope of conventional health technology and behavior interventions. The psychosocial dimension describes social, behavioral, and psychosocial factors that bring about behavior alteration and are acquiescent to intervention. The technological dimension explains behaviors and practices detailed to the WASH technology. It also addresses the continued use and upkeep of latrines [25].

In rural Ethiopia, latrine coverage levels have been well documented both at the national and international levels. However, not many papers address all the factors that drive behavioral change. Data on the multi-level drivers of latrine adoption and use among the rural populace is still inadequate. It is against this background that this article aims to explore the barriers to

the adoption and utilization of improved latrine facilities in rural Ethiopia using the IBM--WASH approach.

## Methods and materials

This study was conducted in the Southern Nations, Nationalities, and Peoples' Region, one of Ethiopia's ten federal regional states, from March to April 2019. Wonago district, located 94 kilometers from the regional capital is situated at 6˚ 19' 00" north latitude and 38˚ 16' 00' east longitude. The projected population of Wonago district in 2018, was 156,274 out of which 75,637(48.4%) were male and 80,637 (51.6%) were female with a population density of 726 people per square kilometer.

The district is approximately 248 km$^2$ and comprises 17 rural Kebeles (the lowest administrative unit in Ethiopia). A Community-Led Total Sanitation and Hygiene program (CLTSH) has been put into practice in all rural Kebeles since 2013. Two individuals from the health extension program, so-called health extension workers (HEWs), have been assigned per Kebele in the district ever since 2005 and have been offering health education door-to-door.

Five out of the seventeen rural Kebeles were selected at random using lottery sampling. The design of this study is qualitative, and thus, qualitative methodologies, i.e., semi-structured key informant interview (KII) and focus group discussion (FGD) were used for data gathering. Two groups of participants were identified and used in this research, to examine the factors affecting the adoption and use of improved latrine facilities at the household and community level; male and female household heads were recruited from the five selected Kebeles in consultation with local civic leaders and Health Extension Workers (HEWs). To investigate the societal factors guiding the adoption and use of improved latrine facilities, Kebele Wash coordinators and Health Extension Workers were used as key Informants.

The study team comprised of two investigators, three data collectors, and one veteran supervisor. Two days of training was given to the supervisor and the data collectors. In the course of the training, the rationale of the study was made clear to the trainees. Furthermore, specifics of the data collecting tools and the techniques they could use to approach the study partakers were discussed. Also, conventions adopted to transcribe audio-recorded data were made clear to the research assistants.

Three focus group discussions were held in each of the rural Kebeles considered in this study. Each group discussion had 7 members. Participants were recruited based on homogeneity, convenience, and willingness to participate. One group included all-female community members, another all-male community member, and the third group was a mixed group of civic leaders. In addition to the FGDs, two key informant interviews were conducted in each Kebele. Participants of the interviews were purposively selected from, the Kebele WASH coordination office and health extension program (HEP) office (Table 1).

The semi-structured key informant interview (KII) and focus group discussion (FGD) guides used for data collection were developed by the research team following an in-depth review of related literature. The data collection guides revolved around knowledge and perceptions regarding sanitation, open defecation, and benefits and/or barriers to adopting and using a latrine. The discussion guides used with both groups of the study participants were made similar to ensure the comparability of the reactions obtained from them (S1 File). The data collection schedule was agreed upon with both the community leaders and the key informants. When the data collection schedule was set, an accord was also reached with the district health bureau and civic leaders to hold the FGDs and the key informant interviews in private to ensure the discretion of the participants. Safeguarding the privacy of participants was needed

**Table 1. Eligibility criteria for the focus groups and key informant interviews, Wonago district, Southern Ethiopia, 2019.**

| Participant category | Inclusion criteria | Exclusion criteria |
|---|---|---|
| Adult FGD | Residents of the study area who are >18years of age | Cognitive impairment and other conditions (e.g. anxiety, impaired hearing, or reduced functional ability representing a severe challenge to group participation and dynamics). |
| | Residents who provide informed consent | |
| Community Leaders' FGD | Residents who are Influencers within the community | |
| Key informants | Personnel responsible for WASH-related duties in the district | District public servants not involved in CLTSH, WASH, and health extension program activities |
| | Personnel in leadership positions of the district WASH and Health extension office | |

to boost the participants' genuine contribution and share as sources of the data needed for the study.

The purpose of the study was clarified to the participants before they signed the informed consent forms. Moreover, the participants were also given confirmation of the research team's assurance to uphold privacy and conceal the exact sources of the opinions brought up. Both the FGDs and the KIIs were conducted until the data reached saturation–a point at which recurring patterns became apparent in the participants' accounts. Both the focus group discussions and the interviews were conducted in Amharic, as that was the lingua-franca used among the study participants and the researchers. Apart from the study participants, every interview and FGD session consisted of a moderator/interviewer, note-taker, and an observer. Each discussion and the interview held with the study participants were audio-recorded. The recording was made with the assent of the study participants.

Open code software 4.03 was used for content analysis, which followed a thematic outline. First, the data collected in Amharic was transcribed verbatim and then translated into English. Translation of the transcript was made by an impartial English language lecturer who had a Doctorate of Philosophy (Ph.D.) in Teaching English as a Foreign Language. Also, the data was analyzed after weighing the original transcript (the Amharic version) against its translated version (the English language version). This guaranteed that there were no inconsistencies in the words, meanings, and subject matter of the items.

Subsequently, the authors separately reviewed the transcripts numerous times to make themselves more acquainted with the content ahead of the process of sorting, coding, and theme identification. Preliminary descriptive codes were based on data immersion with a transcript from each group (females, males, civic leaders, and key informants). Codes were then organized into the five levels and three dimensions of the IBM-WASH model. Initially, the authors developed themes separately. Soon after, the themes that could address the research questions developed from the researchers' exhaustive study of the individually cultivated themes. The thematic categories were further scrutinized to ascertain whether they fit the concepts, propositions, and theories under consideration (IBM-WASH). In reporting the data, relevant verbatim quotes are reported to support the interpretation of the data.

Ethical clearance was obtained from Wachemo University College of Medicine and Health Science's Institutional Review Board (IRB). In advance of data collection, an approval letter was obtained from the Southern Nations, Nationalities and Peoples' Regional Health Bureau, and Wonago District Health Office. Moreover, partaking of respondents was based on their complete agreement.

# Result

## Socio-demographic characteristics of participants

A total of 105 people participated in fifteen focus groups. The mean age of the respondents was 39.9 ± 10.8 and ranged from 18–78. Regarding schooling, 66 of the participants (61.3 percent) had no formal education. In addition, in the all-female focus group, 20% were engaged in farming as a means of earning a livelihood compared to 79.9% in the all-male focus groups (Table 2).

Alongside the focus group discussions, key informant interviews were conducted with one Kebele WASH coordination officer and one health extension program (HEP) officer in each of the five Kebeles. The WASH coordination officers included environmental and public health professionals who were in charge of coordinating water, sanitation, and hygiene activities among public and private stakeholders in the Kebeles. The Health extension program officers were responsible for giving health and health-related door-to-door education to the residents of the study area. Table 3 presents the socio-demographic characteristics of the key informants.

## Barriers to the adoption and use of improved sanitation facilities

The goal of the present study was to explore barriers to the adoption and utilization of improved latrine facilities among the rural residents of Wonago district in the Southern Nations, Nationalities, and Peoples' Region of Ethiopia. The data gathered and analyzed in response to the research concern shed light on some important factors that hampered the adoption and utilization of such facilities among this community. We first present the

**Table 2.  Focus group participants' demographic characteristics, Wonago district, Southern Ethiopia, 2019.**

| Variables | Females FGD (n = 35) | Males FGD (n = 35) | Community Leaders (n = 35) | Total (n = 105) |
|---|---|---|---|---|
| **Sex** | | | | |
| Male | 0 | 35 (100%) | 16 (45.7%) | 51 (48.6%) |
| Female | 35 (100%) | 0 | 19 (54.3%) | 54 (51.4%) |
| **Marital status** | | | | |
| Married | 19 (54.2%) | 28 (80%) | 17 (48.6%) | 64 (60.9%) |
| Single (unwed, divorced) | 13 (37.1%) | 6 (17.2%) | 9 (25.7%) | 28 (26.6%) |
| Widowed | 3 (8.7%) | 1 (2.8%) | 9 (25.7%) | 13 (12.5%) |
| **Age** | | | | |
| 18–20 | 4 (11.4%) | 3 (8.7%) | 0 | 7 (6.6%) |
| 21–29 | 9 (25.7%) | 14 (40%) | 5 (14.3%) | 28 (26.6%) |
| 30–39 | 10 (28.6%) | 5 (14.3%) | 5 (14.3%) | 20 (19.3%) |
| 40–49 | 4 (11.4%) | 4 (11.5) | 8 (22.8%) | 16 (15.2%) |
| ≥50 | 8 (22.8%) | 9(25.5%) | 17 (48.6%) | 34 (32.3%) |
| **Education** | | | | |
| No Formal Education | 24 (68.7%) | 22 (62.8%) | 20 (57.2%) | 66 (61.3%) |
| Primary Education | 8 (22.8%) | 10 (28.7%) | 13 (37.1%) | 31 (29.5%) |
| Secondary Education | 3 (8.5%) | 3 (8.5%) | 2 (5.7%) | 8 (7.6%) |
| **Occupation** | | | | |
| Farming | 7 (20%) | 28 (79.9%) | 29 (82.9%) | 64 (60.9%) |
| Government employee | 4 (11.4%) | 1 (2.9%) | 0 | 5 (4.7%) |
| Micro Enterprise owner | 6 (17.1%) | 1 (2.9%) | 2 (5.7%) | 9 (8.5%) |
| Daily laborer/informal worker | 2 (5.7%) | 5 (14.3%) | 0 | 7 (6.6%) |
| Unemployed | 16 (45.8%) | 0 | 4 (11.4%) | 20 (19.3%) |

**Table 3. Key informants' demographic characteristics, Wonago district, Southern Ethiopia, 2019.**

| Variables | Frequency (n = 10) | Percentage |
|---|---|---|
| **Sex** | | |
| Male | 3 | 30.0% |
| Female | 7 | 70.0%% |
| **Educational Status** | | |
| Diploma Level | 4 | 40.0% |
| Undergraduate Degree | 5 | 50.0% |
| Graduate Degree | 1 | 10.0% |
| **Occupation** | | |
| WASH coordinator | 5 | 50.0% |
| Health Extension Worker | 5 | 50.0% |

important factors of the Contextual dimension of the IBM-WASH in detail. We then present the factors of the Psychosocial and Technology dimensions.

**Contextual dimension.** *Individual factors.* Many individual-level barriers concerning latrine adoption and use were expressed during the focus groups and interviews. Gender, educational status, and personal preference for using the field were repeatedly raised in the participants' testimonies of factors that impede the adoption and consistent use of improved latrines. Gender disparities in sanitation preference were discoursed in detail. Participants noted that women time and again favor open defecation to using a latrine for an array of reasons including smell, fear, and for the reason that it was perceived as strange.

*I get scared of it; there is something scary with a latrine. It's dark and gloomy. I have no choice but to go out (openly defecate) before the morning darkness disappears* (Kebele 4, FGD, Female 62 years old).

Data from key informants tend to agree with FGD participants' data.

*When we go door-to-door to educate people about the benefits of adopting an improved sanitation facility we usually come across people who prefer to openly defecate. They say to us 'what a nuisance the smell would be. . . it feels like we empty the bowels in our beds. We can walk to the fields and do our business there' so yeah, I do agree personal preferences have a big role to play.* (35 years old Male, Health extension worker)

*Some of our older, less educated community members tend to equate adopting and utilizing improved sanitation facilities as being out of tune with the land that feeds them; they think of it as slapping the earth that cares for them.* (45 years old Male, WASH expert)

*Household-level factors.* Another significant piece of information sought in this study was the household-level contextual factors that affect latrine adoption and utilization. Many participants expressed that the adoption, sustainability, and use of improved latrines were influenced by the ability of households to build a new latrine. The researchers endeavored much to understand what *ability* meant. Later, it came to be clear that ability referred to the division of labor within the household and the availability of space on the premises. Participants reported that most non-adopter households are from small spaced or female-headed households.

*Take a look at misses A. She does not have a latrine. She is destitute and she doesn't have the ability. She is a widow with no children. Do you believe she can dig the soil on her own; by*

*herself? Even if we force them to build a latrine can they? These folks require help.* (Kebele 2, FGD, 42 years old Male)

Limited availability of space on the premises owing to a tight spaced compound or limited usable land as a result of the repeated substitution of collapsed latrines or latrines with full collection tanks was also identified as one of the barriers for sustainable latrine adoption and use.

*Our family is too big; we don't even have enough bedrooms. Where can we find the plot to construct a latrine?* (Kebele 3, FGD, 40 years old Male)

*All of us in this Kebele is tested. I have been erecting new latrines every year. Now I have run out of available space in my compound. This makes me feel very disheartened. So my family and I have resorted to open defecation.* (Kebele 5, FGD, 35-year-old Male)

*Community-level factors.* Given that many of these rural Kebeles are primarily agrarian in their lifestyle, many of the men and a few of the women go out to their farm fields and work from dawn until dusk. For many, the lack of public latrines at the community level near their farms encouraged open defecation whilst away from the household. As quoted below, one respondent, for example, was noted expressing his discontent with the lack of public latrines.

*When we are on the farm we don't go back to our residence to use the latrine because it is too far. We just go where we are.* (Kebele 3, FGD, 29 years old Male)

Another added

*It is very difficult whenever we are far from home. There aren't any public latrines. It is much more challenging for a woman when she goes to the market. I suffer from bladder pain as a result of holding my urine for a long time. So now I just urinate on the side of the street hidden from public view if I have to.* (Kebele 2, FGD, 59 years old Female)

A similar dissatisfaction, this time with the cleanliness of public latrines, was raised by another participant.

*. . . It is very horrible seeing someone else's dirt before using a latrine. I do not like cleaning up after someone else's dirt. Public latrines are too dirty.* (Kebele 2, FGD, 19 years old Female)

*Societal-level factors.* At the societal level, focus group partakers talked about the correlation between population density, the status of land ownership, and open defecation. Study participants perceived open defecation as more tolerable in less inhabited areas.

*Our homes are not clustered in a single place rather they are reasonably dispersed, so we have sizable fields to empty the bowels.* (Kebele 3, FGD, 39 years old Female)

Most were also quick to point out that, individuals must own the land they live on to construct a latrine, denoting that tenants or short-term residents would be powerless to build their latrines.

*Those who possess land have the right to erect a latrine and utilize it, however, the law doesn't have provisions to allow non-land owners to build a latrine so in more cases than not those who don't own the land, use the ground.* (Kebele 5, FGD, 69 years old Male)

On the other hand, soil stability emerged as a key test for sustained latrine adoption in some rural areas. As detailed by some participants, in some areas, the soil formation is soft, holds in large amounts of water during the wet season, and cracks in the course of the dry season and as a result, a lot of the latrines fail to withstand soil and climatic situations.

*The soil in our environment is black. It retains lots of water when it drizzles, and it cracks every time it gets dry: this topples the latrines easily.* (Kebele 4, FGD, 49 years old Male)

*Psychosocial dimension.* Culture, beliefs, and attitudes too became apparent as a major impediment to the adoption and consistent use of improved latrine facilities. A key allusion from the interviewees and focus group discussants was that there were some in the community who had a negative attitude towards such facilities. One participant had the following to report in connection with this

*Several households in our community believe that erecting latrines is a waste of resources and that using latrines increases the incidence of diarrhea, especially among the very young.* (Kebele 2, FGD, 40 years old Female)

Key informants did seem to agree with some aspects of the participants' characterizations.

*Extensive efforts were made to educate people about the health impacts and the sheer inappropriateness of excreting in the street where there are so many people around. However, there is always some portion of society that doesn't accept education well.* (36 years old Female, HEW)

*Those community members who hold negative attitudes are reluctant to construct sanitation facilities and believe that using latrines is wrong.* (29 years old Female, HEW)

Women discussants also stated that, in some cases, gender-related cultural norms dissuaded latrine use, particularly among female household members.

*Some older people in this community believe that wedded females shouldn't use the same latrines used by the consanguineous male relatives of their husbands.* (Kebele 1, FGD, 50 years old Female)

For some, perceptions of minimal health threat from a child's feces along with concerns of children coming down with the flu from the malodorous latrines or them slipping and falling into the pit of the latrines extenuated the open defecation of their young children.

*Small children falling into the toilet pit when their mother is distracted by work in the house or goes out to the market is not an uncommon story in our Kebele.* (*Kebele 3, FGD, 45years old Male*)

And yet for a few others, age-old beliefs such as excreting on the ground so that one's refuse serves as a fertilizer for the farm were a motivation to practice open defecation.

*. . . Some people go outside the same way we let our chickens, lambs, and goats defecate on the field; to reduce their fertilizer expenses.* (Kebele 1, FGD, 30 years old Male)

*Technological dimension.* In the course of the focus group discussions, several participants cited inconveniences in acquiring materials to construct a latrine as a barrier to accomplishing

latrine ownership. Some latrine non-adopter households also perceived that they have a lesser amount of capacity judged against latrine-adopter households.

*There is a distinction between us and people who own a latrine. They possess trees to build a sturdy latrine. On the other hand, we use hay, grass, or something similar to that to construct our latrines. Our latrines have been caving in every year. Those who are more capable than us in terms of means do not face the same challenges.*

Another key mention made was the issue of the cost of building a latrine. For some, the demand for increased expenditure that comes with constructing an improved sanitation facility is not something desired.

*We feel embarrassed that a large number of our neighbors have latrines and use them, but we don't due to the lack of resources. Open defecation has a price tag of zero.* (Kebele 5, FGD, Male 40 male years old)

*Ha-ha. . .we dwell in a home with leaky roofs. It is ludicrous to use well-built concrete quarters to empty the bowels.* (Kebele 1, FGD, Female 60 years old)

## Discussion

The purpose of this study was to explore the multi-layered factors influencing the adoption and utilization of latrine facilities in the rural Wonago district of Ethiopia. A recent survey of the availability and utilization of improved sanitation facilities unearthed that, while the coverage of improved sanitation facilities in the study area was 27.3%, the number of people who utilized improved sanitation facilities stood at 64.5% [26]. In the study, it was found that the availability and use of improved latrine facilities were correlated with the educational status of household head, receiving any assistance (monetary or material) from non-governmental organizations, ever receiving educational messages about improved latrine facilities, and frequent supervision by a health extension worker. This study was, therefore, intended to further examine the multi-level constraints to the district's adoption and use of toilet facilities. This type of analysis is critical to policy implementation of CLTSH given that behavioral change is recognized as reflecting upon several factors.

The data gathered and analyzed in response to the research concern shed light on multiple levels of contextual, psychosocial, and technology-related factors associated with the community members' noticeably low level of adoption and utilization of improved latrine facilities. In this study, participants discussed contextual predictors of latrine adoption and use in detail. Gender, educational status, and personal preference for using the field were identified as key individual-level contextual factors. Gender differences in sanitation preference were the most discussed in the focus groups. In some of the studied communities, women were reluctant to use latrines for an array of reasons including fear, and the reason that it was perceived as strange.

Another one of the reasons that discouraged some people, particularly women, from using the latrine was the smell. We considered smell a contextual factor, in this case, because even in situations where the techniques to build a latrine that minimizes or eliminates bad odor were put to use many women still complained of odor and shied away. This indicates that sanitation behavior interventions may perhaps need to be leveled differently to women and men. Another possible explanation for this could be that often time women might not have a say over the latrine design, placement, or maintenance and thus they avoid it altogether. It has

been previously reported that most women in the developing world do not have a voice equal to their partners regarding sanitation; even though they are often the principal implementers of sanitation in the household [27].

Similarly, educational status and personal preference were also identified as important barriers to latrine adoption and use. Some preferred open defecation as they perceived it as being more convenient to latrine use. And, for others improved sanitation was viewed as extravagant and out of tune. This finding is in agreement with the findings of several other studies in Africa and Asia which suggested that educational status and personal preference were crucial motivators for open defecation [22–24].

At the household-level, the unavailability of free space on the premises was recognized as an important barrier to achieving latrine ownership and sustained use. It was seen that in some cases, the high population density and the resultant cramming of houses play a part in the lack of available space for latrines. And in others, the cycle of covering up collapsed or filled pits and reconstructing new latrines resulted in people running out of workable spaces for erecting latrines, predisposing them to go back to the practice of open defecation. This finding confirms the findings reported in studies conducted in Malawi and Rwanda [28, 29]. In these studies, the absence of usable land for latrine construction was found to affect both the adoption and utilization of sanitation facilities.

Another household-level predictor of latrine adoption and use was the gender of the household head. Participants were quick to make suggestions that most non-adopter households were from female-headed households. In many female-headed households, low income coupled with the lack of technical expertise or physical ability to dig soil and erect latrines significantly restricts the choices they can make regarding sanitation [30]. This state of affairs might be bettered if financial aid or training on the engineering skills of latrine construction from local and national governments was available.

Participants also discussed community-level contextual predictors of latrine adoption and use, including distance from farming fields to homes, and uncleanness of shared facilities. Some didn't have access to latrines while working on their farmlands or going to nearby towns while others with access to public or shared latrines were still compelled to openly defecate because of issues related to the long queues and lack of hygiene. In this context, it is important to mention a study conducted in Nepal which reported on the phenomenon of long queues and untidy public latrines. This, according to the finding of the study, may lead to open defecation by compulsion [20].

At the societal level, soil stability, population density, and status of land ownership were identified as the key contextual barriers to sustained adoption and use of improved latrines. For some communities, constructing latrines in unstable soil conditions was found to be taxing. Previous studies have identified climate and soil conditions as being significant determinants of latrine adoption and use [31]. The frequent collapse of latrines due to poor soil conditions is a significant de-motivation for latrine adoption and use [32].

In the course of the interviews, key informants affirmed that there were community-level laws that forbade community members from practicing open defecation. Then again, many participants remarked that in some communities, homes were far away from each other and there were "vast fields" to defecate in. Putting safety laws in place is a matter that requires intelligent and thoughtful preparations. This requires the involvement of the whole community including in the creation and implementation of effective and current safety practices [33].

Another area where the lack of local political commitment was demonstrated is on the issue of land ownership. In Ethiopia, individuals must possess the land they want to construct a latrine on. Tenants or short-term residents are unable to build latrines on their own and would be liable if landlords take legal action for the defacement of property. Until such time

comes where the legislation favors non-land owners in the community, the national policy and sanitation strategy should consider the promotion of suitable latrine options based on societal circumstances, including land ownership.

Concerning the psychosocial element, several participants talked about cultural norms, beliefs, habits, and attitudes that were impediments to latrine adoption and use. In particular, participants stated that several community members recognized that latrines could be beneficial for well-being but still elected not to adopt them. Some community members even erected latrines for demonstration but then did not utilize them. And, in some cases, gender-related cultural norms dissuaded latrine use, particularly among female household members. In terms of developing sanitation interventions to strengthen latrine availability and utilization in populations like these, our study partakers denote that merely schooling communities on the health advantages of latrines may not be adequate to elicit behavior change.

Similarly, the perception of latrines posing a risk to children's safety as a factor may affect the adoption and utilization of latrines [34]. This was also found to be true in this study. Many FGD participants in the present study stated that their latrines were generally not lined with bricks and vulnerable to collapse. This places children in danger. As a result, most households tend to discourage children from using latrines for concerns that they might fall in. likewise, for some FGD participants, perceptions of minimal health threat from children's feces extenuated the open defecation of their young children. Studies in India and Cambodia reported similar results in terms of people's perception of child-feces management [35, 36].

A couple of technological elements were discussed as barriers to latrine adoption and use. Some participants cited problems in procuring materials to build a latrine or the price tag of constructing a latrine as a barrier. Ethiopia at present encourages a simple pit latrine with a slab made from locally obtainable materials [37]. Even though the use of locally obtainable materials has been advocated for the sustainability of any embraced technology, the present study revealed that latrines fell short of being sustained as a result of the use of poor quality local materials. This finding confirms the findings reported in studies conducted in Malawi and Rwanda [29, 38]. In these studies, poor quality of locally available building materials was found to affect the adoption and utilization of sanitation facilities.

Similarly, the cost of building a latrine played a marked constraining role. For some, the demand for increased expenditure that comes with constructing an improved sanitation facility is not something desired. In many cases, the incapacity to shell out for labor and material costs dispirited some households from constructing new latrines or replacing their filled ones. Notwithstanding the Ethiopian sanitation policy, that stipulates zero subsidies for household latrines [39]. Evidence shows that public subsidy has succeeded in other nations [40]. This signifies that some flexibility may be needed vis-à-vis latrine subsidy in Ethiopia.

## Limitations

Our findings must be entertained in the context of several limitations. First, it should be recognized that the study was done among a very homogenous populace with a similar ethnic and religious make-up. The study's discoveries can only be generalized for a principally rural context, apart from factors linked to soil and environmental circumstances, which may apply to locales with a similar context. Further, individual-level behaviors and injunctive norms can be challenging to get from FGDs as people may provide socially preferred responses more than the responses that reflect their actual experiences. Nevertheless, efforts were made to reduce social desirability by making sure only study participants were in attendance in the course of the data collection.

The effectiveness of strategies to strengthen WASH activities depends on the capacity of the individual, household, society, and institutional levels to facilitate and sustain behavior change. The IBM-WASH framework has emerged in response to this role of interventions, but it suffers from a number of limitations. First, within the IBM-WASH framework, rigorous measurement of determinants and the application of measurement theory are largely missing. Several studies have focused on the validity and reliability of self-reported behavioral findings in both domestic and institutional settings, and there is a consensus that alternative approaches are needed to evaluate behavioral outcomes [41]. Second, applying the IBM-WASH framework, we came across several overlaps between the groupings of psychosocial, technological, and contextual dimensions. For example, the smell of latrines can be categorized under technology or the contextual category. Finally, although we were able to categorize individual-level to societal-level factors into the contextual dimension, we were unable to stick to this categorization for the psychosocial and technological dimensions. Apart from these limitations, the study possesses important strengths. The aims of the study are appropriately addressed through a qualitative method, and the study delivers a full and deep assessment of barriers to the adoption and utilization of improved sanitation facilities by taking account of the viewpoints of the members of the community as well as WASH and health extension officers.

## Conclusion

Our findings reveal that contextual, psychosocial, and technological factors influence the adoption and use of improved latrines at several aggregate levels. The use of the IBM-WASH approach helped to achieve a better understanding of these factors concurrently. Providing funding opportunities for the underprivileged and offering training on the engineering skills of latrine construction at the community level based on the contextual soil circumstances could expand the sanitation coverage and use. Moreover, constructing latrines in a manner that is more appealing to women could also further enhance the adoption and use of improved latrines in this community. Finally, taking into account the variability in motivations for adopting and using latrines among our study in Ethiopia and other studies, we implore public health experts to recognize behaviors and norms in their target communities in advance of implementing sanitation interventions. For future efforts, to better represent all rural residents in the Southern Nations, Nationalities & Peoples' Regional State, or in the whole country, a more structured sampling plan is suggested. Also, the use of individual interviews on top of focus groups may better identify personal behaviors and injunctive norms in this population.

## Supporting information

**S1 File. Topic guide for focus group discussions and in-depth discussions.**
(DOCX)

## Acknowledgments

The authors want to acknowledge the respondents, data collectors, and all who were instrumental in the research process.

## Author Contributions

**Conceptualization:** Aiggan Tamene.

**Data curation:** Aiggan Tamene.

**Formal analysis:** Aiggan Tamene.

**Investigation:** Aiggan Tamene.

**Methodology:** Aiggan Tamene.

**Resources:** Aiggan Tamene.

**Supervision:** Abel Afework.

**Validation:** Aiggan Tamene.

**Writing – original draft:** Aiggan Tamene, Abel Afework.

**Writing – review & editing:** Aiggan Tamene.

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
