## [Decision Letter · Decision Letter 0]

25 Nov 2020

PONE-D-20-32012

Exploring Barriers to the Adoption and Utilization of Improved Sanitation Facilities in rural Ethiopia: An Integrated Behavioral Model for Water, Sanitation and Hygiene (IBM-WASH) Approach

PLOS ONE

Dear Dr. Tamene,

Thank you for submitting your manuscript to PLOS ONE. After careful consideration, we feel that it has merit but does not fully meet PLOS ONE’s publication criteria as it currently stands. Therefore, we invite you to submit a revised version of the manuscript that addresses the points raised during the review process.

Please address the reviewers' suggestions for minor revisions, below.

We look forward to receiving your revised manuscript.

Kind regards,

Susan Horton

Academic Editor

PLOS ONE

Journal Requirements:

2.  Please provide additional information regarding the participant eligibility criteria for the two groups.

3. Thank you for stating the following in the Funding Statement section of your manuscript:

"The research was performed as part of the employment of the authors at Wachemo University

and Dilla University. Only the authors were involved in the manuscript writing, editing,

approval, or decision to publish."

Reviewers' comments:

Reviewer's Responses to Questions

**Comments to the Author**

1. Is the manuscript technically sound, and do the data support the conclusions?

Reviewer #1: Yes

Reviewer #2: Yes

2. Has the statistical analysis been performed appropriately and rigorously? 

Reviewer #1: N/A

Reviewer #2: N/A

3. Have the authors made all data underlying the findings in their manuscript fully available?

Reviewer #1: No

Reviewer #2: No

4. Is the manuscript presented in an intelligible fashion and written in standard English?

Reviewer #1: Yes

Reviewer #2: Yes

5. Review Comments to the Author

Reviewer #1: This is a very solid paper, with clear methods and relevant results that provide great qualitative insights into understanding behavioral change in latrine-use within the context of a rural district in Ethiopia. A few comments I hope the authors will consider follow.

In the introduction and then again in the conclusion, you state that – for example on pg 19 – “This study presents the barriers (perceived and actual) to the adoption and utilization of

improved sanitation facilities which were not sufficiently addressed in the existing quantitative

literature.” This critique of existing literature is a little misleading. First it aims only at quantitative literatures in the conclusion. Yet one can argue that it is not that existing literatures do not sufficiently address drivers or barriers to adoption as a whole, but that existing studies do not typically combine all dimensions of the analysis into one study – this is a methodological issue if any. In other words, quantitative studies may simply be choosing to understand the weight and scale of a particular vector or driver of behavioral change; as opposed to indicating that all other vectors are not important, they are testing a specific hypothesis about one vector. Your statements make it seem like all studies must study all aspects in one analysis, otherwise they are not useful. Is that really what you wish to say and what is the basis of such an evaluation? While I very much sympathize with your methodological approach, I fear that the paper here discounts too rapidly the contributions of other methods of analysis outside of IBM-WASH. If that is the point of the analysis and argument in this paper, then you should add more comparison of different studies and their methods, and a clear metric of evaluation of those studies. Alternatively, I’d recommend simply indicating that not many papers address all factors driving behavioral change, and that this type of analysis is critical to policy implementation of CLTSH given that behavioral change is recognized as reflecting upon several factors.

It seems your discussion is really centered on latrine use, but the title and introduction of your article point to the wider portfolio of improved sanitation. I would suggest a correction to both the title of the piece, and a more specific concentration on latrine typologies, recommendations therein, and latrine use in the introduction, and perhaps reference to open defecation, given the connections therein.

The drop from 80% in 2000 to 27% in 2015 in Ethiopia with regard to open defecation is a rather tremendous success story. How does Wonago district in the Southern Nations perform on this metric? Can you speak to this in the introduction? What do other studies indicate drove that improvement in rural parts of Ethiopia, and why, if absent in Wonago, was it not present in the case you are discussing? These seem pertinent points that could be better addressed in the introduction, adding deeper nuance to the presentation of your case and moving away from the more generic discussion of sanitation improvement policies and the SDGs currently there.

On pg 15, you write: A recent survey on the availability and utilization of improved sanitation facilities unearthed that, while the coverage of improved sanitation facilities in the study area was 27.3%, the number of people who utilized improved sanitation facilities stood at 64.5% [22].

Can you tell us more about this finding and study, particularly given its focus on the Wonago district? The findings you report explain very little from the study you cite, yet it would seem rather relevant to discuss it further in your paper.

Reviewer #2: This paper explores the barriers to adoption and use of improved sanitation facilities in rural Ethiopia. 15 focus groups plus 10 key informant interviews across 3 rural communities. Barriers to adoption and use were categorized into contextual factors (gender; education; personal preference; limited space; population density; status of land ownership); psychosocial factors (culture; beliefs, attitudes and perceptions of minimal health threat from children’s faeces), and technological factors (inconvenience in acquiring materials; costs of constructing the latrine) using the IBM WASH framework.

Conclusion – communities in developing nations require money and training; and, those who provide money and training require cultural sensitivity to local customs and norms/mores.

This is a very thorough reporting of a very comprehensive study of a very important issue. Well presented, well written. I only have one comment which I think is very important but because there is only one issue, in my view, I suggested minor review. My issue is: the IBM-WASH model is applied relatively uncritically. That is, I fully appreciate its development is well grounded in the literature and it was developed by leading WASH scholars, but no framework is perfect and I wonder if the authors might address a this issue somewhat in the discussion or conclusions of their paper. For example, an alternative framework might use context (e.g., population density; land ownership), composition (e.g., gender, education, personal preference) and collective (e.g., cultural norms such as the attitudes toward children's faeces). The authors DO hint a bit at some of the shortcomings of the framework in the limitations, but it's more implicit than explicit. To be clear, I am not suggesting at all that the use of the IBM-WASH framework is incorrect or that the authors should switch to an alternative framework. Rather, I would just like them to engage with the framework they chose a little bit more through a critical lense regarding its strengths as well as its weaknesses.

6. PLOS authors have the option to publish the peer review history of their article (what does this mean?). If published, this will include your full peer review and any attached files.

Reviewer #1: No

Reviewer #2: No

---

## [Author Response · Author response to Decision Letter 0]

30 Nov 2020

Authors’ Response to the Academic Editor and Reviewers

Dear: Susan Horton, Academic Editor

We thank you and the reviewers for your thorough reading and constructive criticism of our manuscript and for the opportunity to revise and resubmit. We are pleased to submit the revised research article entitled “Exploring Barriers to the Adoption and Utilization of Improved Latrine Facilities in rural Ethiopia: An Integrated Behavioral Model for Water, Sanitation and Hygiene (IBM-WASH) Approach” for your consideration in PLOS ONE. On the following pages, you will find our response to the reviewers’ comments. On behalf of my co-author, I thank you again for your consideration of this resubmission. We appreciate your time and look forward to your response.

Sincerely, 

Aiggan Tamene

[Corresponding author]

Point by point response to the reviewers (Reviewer 1)

We would like to thank the reviewer for his/her time and effort made to review our manuscript in detail. We believe that the comments have identified important areas which required improvement. After completion of the suggested edits, the revised manuscript has benefitted from an improvement in the overall presentation and clarity. We have used Microsoft review track change in the document to indicate any changes in word, phrase or sentence. 

Comment: In the introduction and then again in the conclusion, you state that – for example on pg 19 – “This study presents the barriers (perceived and actual) to the adoption and utilization of

improved sanitation facilities which were not sufficiently addressed in the existing quantitative

literature.” This critique of existing literature is a little misleading. First it aims only at quantitative literatures in the conclusion. Yet one can argue that it is not that existing literatures do not sufficiently address drivers or barriers to adoption as a whole, but that existing studies do not typically combine all dimensions of the analysis into one study – this is a methodological issue if any. In other words, quantitative studies may simply be choosing to understand the weight and scale of a particular vector or driver of behavioral change; as opposed to indicating that all other vectors are not important, they are testing a specific hypothesis about one vector. Your statements make it seem like all studies must study all aspects in one analysis, otherwise they are not useful. Is that really what you wish to say and what is the basis of such an evaluation? While I very much sympathize with your methodological approach, I fear that the paper here discounts too rapidly the contributions of other methods of analysis outside of IBM-WASH. If that is the point of the analysis and argument in this paper, then you should add more comparison of different studies and their methods, and a clear metric of evaluation of those studies. Alternatively, I’d recommend simply indicating that not many papers address all factors driving behavioral change, and that this type of analysis is critical to policy implementation of CLTSH given that behavioral change is recognized as reflecting upon several factors.

Response - As the esteemed reviewer rightly pointed out because of differences in their methodological approach different studies reach varying conclusion on a single problem area. We want to emphasize that it was not the authors’ intention to disregard the findings of other studies (quantitative and qualitative) rather in our quest to highlight that the IBM-WASH approach helped us to achieve a better understanding of the multi-level drivers of latrine adoption and use concurrently, we regrettably made assumptions that other studies were lacking in their approach. As a result, rather than risking the advent discounting the contributions other papers have made we have taken the recommendation given by the esteemed reviewer with full heart and removed any insinuations to the need for all studies to asses all aspects of latrine adoption and use in one analysis and we have re-worded the conclusion and introduction sections to indicate that “not many papers address all factors driving behavioral change, and that this type of analysis is critical to policy implementation of CLTSH given that behavioral change is recognized as reflecting upon several factors.”

Comment: It seems your discussion is really centered on latrine use, but the title and introduction of your article point to the wider portfolio of improved sanitation. I would suggest a correction to both the title of the piece, and a more specific concentration on latrine typologies, recommendations therein, and latrine use in the introduction, and perhaps reference to open defecation, given the connections therein.

Response: We thank the reviewer for his/her astute observation. The reviewer rightfully pointed out that the title of the piece and the contents within it need to align to the fullest extent. Accordingly so, the authors have made corrections to the title of the piece and more specific concentration on latrine typologies, latrine recommendations, and latrine use in the introduction, 

Comment: The drop from 80% in 2000 to 27% in 2015 in Ethiopia with regard to open defecation is a rather tremendous success story. How does Wonago district in the Southern Nations perform on this metric? Can you speak to this in the introduction? What do other studies indicate drove that improvement in rural parts of Ethiopia, and why, if absent in Wonago, was it not present in the case you are discussing? 

Response: As the reviewer rightfully pointed out there was a gap in informing readers about the drivers of the decline in open defecation within the country. In the revised version of the manuscript however, there was a serious attempt to move away from the more generic discussion of sanitation improvement policies and the SDGs and address the factors that have largely allowed this significant achievement nationally. Similarly, within the Southern Nations, Nationalities, and Peoples’ Region of Ethiopia, the profile of latrine adoption status and the comparison of the Wonago district to the national figures and the reasons why Wonago district has not kept pace with the national increase in latrine adoption and utilization were included in the current version. 

Comment: On pg 15, you write: A recent survey on the availability and utilization of improved sanitation facilities unearthed that, while the coverage of improved sanitation facilities in the study area was 27.3%, the number of people who utilized improved sanitation facilities stood at 64.5%. Can you tell us more about this finding and study, particularly given its focus on the Wonago district? The findings you report explain very little from the study you cite, yet it would seem rather relevant to discuss it further in your paper.

Response: Thank you for your pertinent comment. In the present revision of the manuscript, we have placed a strong emphasis on the prior quantitative study conducted in the same study area. We tried to present the findings of the study and discuss the factors correlated with the outcome. In addition, we explained that the intention our present study to further examine the multi-level constraints to the district's adoption and use of toilet facilities. We also emphasized that this type of analysis is critical to policy implementation of CLTSH given that behavioral change is recognized as reflecting upon several factors.

Point by point response to the reviewers (Reviewer 2)

We would like to thank the reviewer for his/her time and effort made to review our manuscript in detail. We believe that the comments have identified important areas which required improvement. After completion of the suggested edits, the revised manuscript has benefitted from an improvement in the overall presentation and clarity. We have used Microsoft review track change in the document to indicate any changes in word, phrase or sentence. 

Comment: My issue is: the IBM-WASH model is applied relatively uncritically. That is, I fully appreciate its development is well grounded in the literature and it was developed by leading WASH scholars, but no framework is perfect and I wonder if the authors might address a. this issue somewhat in the discussion or conclusions of their paper. For example, an alternative framework might use context (e.g., population density; land ownership), composition (e.g., gender, education, personal preference) and collective (e.g., cultural norms such as the attitudes toward children's feces). The authors DO hint a bit at some of the shortcomings of the framework in the limitations, but it's more implicit than explicit. To be clear, I am not suggesting at all that the use of the IBM-WASH framework is incorrect or that the authors should switch to an alternative framework. Rather, I would just like them to engage with the framework they chose a little bit more through a critical lens regarding its strengths as well as its weaknesses.

Response: We would like to appreciate you for your in-depth examination of our manuscript. As the reviewer rightfully pointed out, a number of WASH-specific models and frameworks exist, yet with some limitations. When reviewing WASH-specific theoretical models to apply in the present study, the authors came to a conclusion that the IBM-WASH model was ideal in our setting as it aims to provide both a conceptual and practical tool for improving our understanding and evaluation of the multi-level multi-dimensional factors that influence water, sanitation, and hygiene practices in infrastructure constrained settings. However, in the revised manuscript we have tried to critically engage with the manuscript and delve deeper into the limitations that came with this specific model. We also tried to outline future research priorities needed to advance our understanding of the sustained adoption of water, sanitation, and hygiene technologies and practices.

END ________________________________________

Comment: PLOS authors have the option to publish the peer review history of their article (what does this mean?). If published, this will include your full peer review and any attached files. Response: No 

We hope the esteemed reviewers can see that the revised manuscript has tried to accommodate the much deserved constructive criticisms to the fullest extent of our abilities. 

Best Regards,

Thank you!

---

## [Editor Report · Decision Letter 1]

23 Dec 2020

PONE-D-20-32012R1

Exploring Barriers to the Adoption and Utilization of Improved Sanitation Facilities in rural Ethiopia: An Integrated Behavioral Model for Water, Sanitation and Hygiene (IBM-WASH) Approach

PLOS ONE

Dear Dr. Tamene,

Thank you for submitting your manuscript to PLOS ONE. After careful consideration, we feel that it has merit but does not fully meet PLOS ONE’s publication criteria as it currently stands. Therefore, we invite you to submit a revised version of the manuscript that addresses the points raised during the review process.

I am satisfied that you have responded appropriately to the reviewers' comments. I would request one further small change. Best practice for qualitative research is to follow the SRQR checklist. The manuscript mostly fulfils those criteria, with the exception of an explanation (which can be brief) as to how the FGD participants were recruited. Please add a couple of sentences to explain this.

We look forward to receiving your revised manuscript.

Kind regards,

Susan Horton

Academic Editor

PLOS ONE

Additional Editor Comments (if provided):

Thank you for responding appropriately to the reviewers' comments. I would ask you to make one small additional further change, namely, just prior to Table 1, include a couple of sentences explaining how the participants were recruited for the two FGD.

---

## [Author Response · Author response to Decision Letter 1]

24 Dec 2020

As the esteemed Editor rightly pointed out, an explanation of how the study participants were recruited was missing from the manuscript. We apologize for not addressing it after the first comment. We misunderstood the Editor’s comments and only added the eligibility criteria used in the present study. We want to emphasize that it was not the authors’ intention to not address such a critical revision to our manuscript. We used purposive sampling using local contacts as we reached a conclusion that random sampling when selecting FGD participants. Might compromise the findings as the groups need to be homogenous and all individual members need to share the characteristic relevant to our information needs in order for us to get the most out of the discussion

---

## [Editor Report · Decision Letter 2]

26 Dec 2020

Exploring Barriers to the Adoption and Utilization of Improved Sanitation Facilities in rural Ethiopia: An Integrated Behavioral Model for Water, Sanitation and Hygiene (IBM-WASH) Approach

PONE-D-20-32012R2

Dear Dr. Tamene,

We’re pleased to inform you that your manuscript has been judged scientifically suitable for publication and will be formally accepted for publication once it meets all outstanding technical requirements.

Kind regards,

Susan Horton

Academic Editor

PLOS ONE

Additional Editor Comments (optional):

Thank you for making the final revisions so quickly. I look forward to seeing the article published.

---

## [Editor Report · Acceptance letter]

2 Jan 2021

PONE-D-20-32012R2 

Exploring Barriers to the Adoption and Utilization of Improved Latrine Facilities in rural Ethiopia: An Integrated Behavioral Model for Water, Sanitation and Hygiene (IBM-WASH) Approach 

Dear Dr. Tamene:

I'm pleased to inform you that your manuscript has been deemed suitable for publication in PLOS ONE. Congratulations! Your manuscript is now with our production department. 

Kind regards, 

on behalf of

Dr. Susan Horton 

Academic Editor

PLOS ONE